# Autistic and Catatonic Spectrum Symptoms in Patients with Borderline Personality Disorder

**DOI:** 10.3390/brainsci13081175

**Published:** 2023-08-07

**Authors:** Liliana Dell’Osso, Giulia Amatori, Ivan Mirko Cremone, Enrico Massimetti, Benedetta Nardi, Davide Gravina, Francesca Benedetti, Maria Rosaria Anna Muscatello, Maurizio Pompili, Pierluigi Politi, Antonio Vita, Mario Maj, Barbara Carpita

**Affiliations:** 1Department of Clinical and Experimental Medicine, University of Pisa, 56126 Pisa, Italy; liliana.dellosso@gmail.com (L.D.); ivan.cremone@gmail.com (I.M.C.); benedetta.nardi@live.it (B.N.); davide.gravina@hotmail.it (D.G.); francesca.benedetti1793@gmail.com (F.B.); barbara.carpita1986@gmail.com (B.C.); 2ASST Bergamo Ovest, SSD Psychiatric Diagnosis and Treatment Service, 24047 Treviglio, Italy; e.massimetti@yahoo.it; 3Department of Biomedical and Dental Sciences and Morphofunctional Imaging, University of Messina, 98124 Messina, Italy; mmuscatello@unime.it; 4Department of Neuroscience, Mental Health and Sense Organs, University of Roma “La Sapienza”, 00185 Roma, Italy; maurizio.pompili@uniroma.it; 5Department of Brain and Behavioral Sciences, University of Pavia, 27100 Pavia, Italy; pierluigi.politi@unipv.it; 6Department of Clinical and Experimental Sciences, University of Brescia, 25123 Brescia, Italy; antonio.vita@unipv.it; 7Department of Psychiatry, University of Naples “Luigi Vanvitelli”, 80138 Naples, Italy; mario.maj@unicampania.it

**Keywords:** autism spectrum disorder, catatonia, autism spectrum, catatonia spectrum, borderline personality disorder

## Abstract

Background: Recent literature has shown that a considerable percentage of patients with severe mental disorders can develop, over time, full-blown or subthreshold catatonia. Some studies corroborate the model of an illness trajectory in which different mental disorders would be arranged along a continuum of severity until the development of catatonia. In such an illness pathway, autistic traits (AT) and borderline personality disorder (BPD) may represent important steps. In order to further explore the association between AT, BPD, and catatonia, the aim of this study was to compare catatonic spectrum symptoms and AT among patients with major depressive disorder (MDD), BPD, and healthy controls (CTL), also evaluating possible predictive dimensions of the different diagnoses. Methods: A total of 90 adults affected by BPD, 90 adults with a diagnosis of MDD, and 90 CTL, homogeneous in terms of gender and age, were recruited from six Italian university departments of psychiatry and assessed with the SCID-5-RV, the Catatonia Spectrum (CS), and the Adult Autism Subthreshold Autism Spectrum (AdAS Spectrum). Results: The total CS score was significantly higher in the BPD and MDD groups than in the CTL group, while the majority of CS domain scores were significantly higher in the BPD group than in the MDD group, which scored significantly higher than the CTL group. The total AdAS Spectrum score and the AdAS Spectrum domain scores were significantly higher in the BPD group than in the MDD group, which in turn scored significantly higher than the CTL group. The CS domains “psychomotor activity” and “impulsivity”, and AdAS Spectrum domains “verbal communication”, “empathy”, and “hyper-/hyporeactivity to sensory input” were associated with the risk of presenting a diagnosis of BPD.

## 1. Introduction

The first description of catatonia dates back to 1874, when German psychiatrist Karl Ludwig Kahlbaum used the term to define an autonomous, cyclical, and progressive nosographic entity characterized by alternating astonishment and excitement [1] and behavioral and motor manifestations (negativism, mutism, stereotypies, mannerisms, automatic obedience, automatisms, impulsivity, and agitation) in combination with cognitive, affective, and neurovegetative symptoms. This definition was challenged and, in part, reformulated by Emil Kraepelin (1856–1926), who in the magnum opus “Kompendium der Psychiatrie”, first published in 1883, distinguished two forms of psychosis: manic depression and dementia praecox, considering catatonia as one of the possible forms of the latter. The Kraepelinian conceptualization of mental disorders exerted an unprecedented influence on the course of psychiatry, becoming the basis for what was reported in the first edition of the Diagnostic and Statistical Manual of Mental Disorders (DSM) [2]. Catatonia thus remained relegated to the realm of psychoses for about a century. Kraepelin and Bleuler’s perspective was maintained in the first four editions of the DSM, until the conceptualization, similar to Kahlbaum’s original, presented by the fifth edition of the manual (DSM-5) [3]. Within the DSM-5 and its text revision (DSM-5-TR) [4], catatonia has been placed in the “Schizophrenia Spectrum and Other Psychotic Disorders” chapter, along with schizophrenia, other psychotic disorders, and schizotypal personality disorder. It is described as a complex neuropsychiatric syndrome characterized by a constellation of psychomotor signs and symptoms developed in the context of numerous pathological conditions, not only psychiatric, but also neurological, toxic, endocrinological, and infectious [3]. Catatonic symptoms are nonspecific and can also be found in other mental disorders. In fact, the DSM-5 does not conceive catatonia as an independent category, but rather as a transnosographic specifier, describing three distinct conditions: “catatonia associated with another mental disorder”, “catatonic disorder due to another medical condition”, and “unspecified catatonia”, a provisional diagnosis to be used when the nature of the underlying disorder is unclear, there is insufficient information, or the criteria for catatonia are not fully met. According to the DSM definition, the category of “unspecified catatonia” also includes subthreshold forms of catatonia, opening the way to the definition of a “catatonic spectrum”. Although the pathophysiology of catatonia is still unclear, there is evidence of a wide variety of underlying disorders that can be associated with the emergence of catatonic signs, including medical and mental disorders. In this regard, full-blown catatonia has been reported to occur in more than 10% of patients with acute psychiatric illnesses [5], in particular among patients with autism spectrum disorder (ASD), and especially in younger ones [6]. According to other authors, overt symptoms of catatonia have been detected in 18% of adolescents admitted to a specialized psychiatric ward due to a variety of mental disorders, such as pervasive developmental disorder, psychosis, disruptive behavior disorder, intellectual disability, and autism spectrum disorder (ASD) [7]. Regarding children with ASD, the literature shows that the occurrence of catatonia is not a rare event [8,9]. In two prevalence studies [10,11], a diagnosis of catatonia was made in 12–17% of a large sample of adolescents and young adults with ASD. ASD and catatonia also share many clinical manifestations such as mutism, echolalia, stereotyped movements, repetitive behaviors, negativism, and arousal. Even from a neurophysiological perspective, the cortical GABA-ergic dysregulation proposed as an etiological model for catatonia has been considered one of the possible alterations of neurotransmission underlying ASD [12]. Autism spectrum has been proposed as a transnosographic dimension [13,14], representing a risk factor for the development of mental disorders [15,16]. This pattern has been observed, over time, in numerous samples of individuals with psychiatric diseases, such as bipolar disorder, borderline personality disorder (BPD), and eating disorder, as well as in nonclinical samples [15,17,18]. Regarding the correlation with borderline personality disorder, numerous clinical observations suggest a link between BPD and ASD, especially in the case of high-functioning autism [19]. Acting out phenomena, alterations in empathy, transient paranoid ideation, feelings of anger, and self-injurious gestures, are indeed common in both disorders. Even at the cognitive level, ASD and BPD share difficulties in reading others’ emotions and alterations in social cognition. [19]. A recent study, moreover, has shown high suicidality in patients with comorbidity between the two disorders, extended also to nonclinical individuals with concurrent autistic and borderline traits [20]. Beyond a pure question of comorbidity, some researchers have hypothesized the existence of common etiological roots, pointing to how a history of traumatic events, in patients characterized by autistic traits, implying social cognition impairment, lower adaptive capacity, ruminative thinking, and sensorial hypersensitivity, may lead to the development of a clinical picture called “complex post-traumatic stress disorder” (cPTSD), clinically very similar to BPD [21]. Even in the context of catatonia, traumatic and highly stressful events could play a significant role in the pathogenesis of the disorder. Shorter and Fink described catatonia as “madness of fear” describing catatonic patients as “overwhelmed by fear, terror and anxiety” distinguished by “rigid muscles, as in the fear response of an animal before a predator” [22]. Well before Shorter and Fink’s brilliant observations, the traumatic origin of catatonia was captured in the clinical pictures of World War I soldiers. The earliest reported work on the Pubmed database related to catatonia dates back to 1921 and describes the case of a young soldier plunged, as a result of psychic trauma experienced on the battlefield, into a state of catatonia, characterized by stupor, mutism, negativism, catalepsy, waxy flexibility, and fixity of gaze, in which he persisted for the duration of four years, until a sudden awakening with almost immediate recovery of premorbid functioning, followed by an unfortunate and rapid relapse into the stupor state from which he had awakened [23]. Other recent studies would seem to support the hypothesis of an etiological contribution of psychic trauma in catatonia [24,25]. Another shared factor between BPD, ASD, and catatonia is the increased suicidality risk [26,27]. In this regard, a recent study found a significant correlation between high catatonic traits and high suicidal tendencies in BPD patients [15]. In a recent validation study of a questionnaire investigating the manifestations of subthreshold catatonia in a sample of patients with major depressive disorder (MDD), BPD, and catatonia, a progressive increase in catatonic traits was observed from healthy controls to patients with MDD to subjects with BPD up to the catatonia group [28]. In another study on three groups of patients with ASD, subthreshold autistic traits, and no autistic traits, catatonic traits appeared to increase in parallel with the increase in autistic traits [16]. In the same study, the group of patients with overt autism spectrum, corresponding to the highest catatonic traits, was found among patients with BPD. All this led to the hypothesis of the existence of a psychopathological trajectory originating from an autistic-type neurodevelopmental disorder and culminating in catatonia, in which BPD would represent a significant stage of severity. In this framework, in order to further explore the association between autistic traits, BPD, and catatonia, the aim of this study was to compare catatonic spectrum features and autistic traits among patients with MDD, BPD, and healthy controls (CTL), also evaluating possible predictive dimensions of the different diagnoses.

## 2. Materials and Methods

### 2.1. Study Sample and Procedures

The total sample included 270 subjects divided into three diagnostic groups, all assessed by trained clinicians according to the DSM-5 diagnostic criteria. The exclusion criteria were as follows: age under 18 years, language or intellectual difficulties affecting the ability to perform assessments, mental disability, poor ability to cooperate, and ongoing psychotic symptoms. More specifically, three groups were identified as follows: 90 subjects with a diagnosis of BPD; 90 subjects with a diagnosis of MDD; and 90 healthy controls with no current or past mental disorders (CTL) and belonging to medical and paramedical staff. All subjects were between 18 and 60 years of age and signed a written informed consent form. The Structured Clinical Interview for DSM-5, Research Version (SCID-5-RV) [29] was used to corroborate the diagnoses of BPD and MDD as well as the absence of mental disorders among CTL. The study was performed in conformity with the Declaration of Helsinki. The Ethics Committee of the Azienda Ospedaliero-Universitaria di Pisa approved all recruitment and evaluation procedures. Eligible subjects provided written informed consent after receiving a complete description of the study and being given the opportunity to ask questions. Subjects received no payment for their participation, according to Italian law.

### 2.2. Measures

The assessment procedures consisted of the SCID-5-RD [29], the Subthreshold Adult Autism Spectrum (AdAS spectrum), and the Catatonia Spectrum (CS). The questionnaires were administered by psychiatrists trained and certified in the use of the instruments.

#### 2.2.1. The Adult Autism Subthreshold Spectrum

The Adult Autism Subthreshold Spectrum (AdAS Spectrum) is a questionnaire developed by Dell’Osso et al. [14] and devised to assess not only full-blown ASD but also the broader spectrum of subthreshold autism, in subjects with normal intelligence and without language impairment across their lifetime. It allows the evaluation of a wide range of clinical and non-clinical traits, and typical and atypical manifestations, including some gender-specific features. The instrument is composed of dichotomous questions, grouped into seven domains: childhood/adolescence, verbal communication, non-verbal communication, empathy, inflexibility and adherence to routine, restricted interests and rumination, and hyper-/hyporeactivity to sensory input. In the validation study [14]. The AdAS Spectrum questionnaire demonstrated excellent reliability and a strong convergent validity with other scales employed in this field, such as the Autism-Spectrum Quotient Test [30] and the Ritvo Autism and Asperger Diagnostic Scale 14-item version [31].

#### 2.2.2. The Catatonia Spectrum

The Catatonia Spectrum (CS) is a self-assessment questionnaire that investigates nuclear, subthreshold, atypical, and partial manifestations of the CS, referred to across the lifespan, divided into domains, and explored with a set of questions. The CS consists of 74 items and is divided into 8 domains: psychomotor activity (stupor), verbal response (mutism), repetitive movements (stereotypes), artificial expressions and actions (mannerisms), oppositivity or poor response to stimuli, response to instructions given from outside (automatic obedience), automatisms, and impulsivity. For each item, there is a dichotomous answer “yes” and “no”. In the validation study [28], the CS questionnaire demonstrated excellent internal consistency and test–retest reliability, and strong convergent validity with alternative dimensional measures of catatonia, such as the Bush–Francis Catatonia Rating Scale [32] and the Bush–Francis Catatonia Screening Instrument [32].

### 2.3. Statistical Analyses

Analyses were performed using Statistical Package for the Social Sciences (SPSS) version 26.0 [33].

The AdAS Spectrum and CS mean total and domain scores reported in the four diagnostic groups were compared through a one-way analysis of variance (ANOVA). The Bonferroni test was used for post hoc comparisons. Multinomial regression was performed to identify the CS and AdAS Spectrum domains more strongly associated with the diagnoses of BPD and MDD, using the CTL group as the reference category. Finally, we used a discriminant function analysis, in which the diagnosis of the patients (BPD, MDD, or CTL) was the dependent variable and the CS and AdAS domains were the independent variables, in order to produce a linear combination of a subset of the predictors that would most efficiently discriminate between the two groups.

## 3. Results

The three diagnostic groups were sufficiently homogeneous in terms of age and gender.

The BPD group included subjects with a mean age of 36.71 ± 15.12 years and consisted of 26 (28.9%) males and 64 (71.1%) females. The MDD subjects had a mean age of 38.18 ± 10.03 years and the group consisted of 27 (30.0%) males and 63 (70.0%) females. The group of CTL subjects had a mean age of 34.94 ± 8.156 years and consisted of 27 (30.0%) males and 63 (70.0%) females.

As shown by ANOVA and post hoc comparisons, the total CS score was significantly higher in the BPD and MDD groups than in the CTL group (*p* < 0.001), as were the scores of the CS psychomotor activity, verbal response, artificial expressions, and response to instructions domains. All remaining CS domains showed significantly higher scores in the BPD group than in the MDD group, which in turn scored significantly higher than CTLs (*p* < 0.05). The total AdAS Spectrum score was significantly higher in the BPD group than in the MDD group, which in turn reported significantly higher scores than the CTL group (*p* < 0.01), as observed for all questionnaire domains (*p* < 0.05). Comparisons between CS and AdAS Spectrum total and domain scores among the diagnostic groups are reported in Table 1.

Multinomial regression showed that high scores in the CS psychomotor activity (*p* < 0.01) and impulsivity (*p* < 0.01) domains, as well as high scores in the AdAS Spectrum verbal communication (*p* < 0.015), empathy (*p* < 0.016), and hyper-/hyporeactivity to sensory input (*p* < 0.015) domains, were statistically predictive factors of a BPD diagnosis, while the CS domains artificial expressions (*p* < 0.016) and oppositivity (*p* < 0.010) seem to play a protective role. Regarding the MDD diagnosis status, high scores in the CS domain psychomotor activity (*p* < 0.001) and the AdAS Spectrum domain hyper-/hyporeactivity to sensory input (*p* < 0.004) were found to be risk factors, while the CS domains artificial expressions (*p* < 0.017) and oppositivity (*p* < 0.003), and the AdAS Spectrum domain nonverbal communication (*p* < 0.048) seemed to play a protective role (Table 2).

Discriminant analysis showed that the first discriminant function absorbed 84.5% of the variance and was able to discriminate well among the three diagnostic groups BPD, MDD, and CTL (Figure 1). All of the CS and AdAS Spectrum domains seem to have significant weight within the first discriminant function. The highest discriminant values were reported for the CS psychomotor activity and the AdAS Spectrum hyper-/hyporeactivity to sensory input domains (Table 3).

## 4. Discussion

According to our data, the total CS score and all its individual domain scores were higher in the group of patients with BPD disorder than in healthy controls. In addition, the scores of most domains were also higher in the BPD group than those observed in the MDD group. A similar result was also observed with regard to autistic traits since both the total AdAS Spectrum score and individual domain scores were higher in the BPD group than in the other two diagnostic categories. These results suggest the high representation of autistic and catatonic traits, whose correlation was already reported in a previous study [15], in the psychopathological picture of BPD patients, supporting the hypothesis of an illness trajectory between autism and catatonia, in which BPD would represent an extremely significant stage.

In particular, the CS domains related to alterations in psychomotor activity and impulsivity, and the AdAS Spectrum domains related to difficulties in verbal communication and altered sensitivity to stimuli, would seem to be the psychopathological dimensions most closely related to the risk of a BPD diagnosis. This finding is very interesting, considering that catatonic-type psychomotor symptoms such as stupor and excitement, especially in an attenuated subthreshold form, could exhibit a clinical overlap with manifestations of BPD such as severe dissociative states or episodes of behavioral dysregulation. Moreover, the impulsivity domain, originally reported by Kahlbaum for the description of the clinical presentation of catatonia, represents one of the nuclear symptoms provided by the DSM for the diagnosis of BPD. Regarding the AdAS Spectrum domains related to verbal communication difficulties and empathic deficits, it is known from the literature that patients with BPD would be characterized by altered emotional empathy, experiencing higher rates of emotional contagion when emotions are expressed nonverbally, contributing to misunderstandings and inadequate social behavior [34]. There is also evidence of overall expressive language impairment and reduced syntactic and lexical complexity in BPD patients [35]. Finally, regarding the sensory domain, a previous study showed the presence of heightened self-reported reactivity to aversive sounds in patients with BPD [36]. The altered sensory impairment could also be related to the post-traumatic hyperarousal potentially found in individuals with BPD, as this disorder is known to be related to past traumatic experiences. Ultimately, in patients with BPD, altered sensitivity to sensory stimuli could increase the vividness of traumatic memories, enhancing the phenomenon of post-traumatic re-experiencing and promoting the structuring of post-traumatic disorder. Considering the results reported so far, it would be interesting to investigate the potential usefulness of therapeutic interventions targeting the specific domains mentioned above in improving the clinical picture of patients with BPD. It would also be useful to explore, in future studies, the potential influence of variables such as comorbidities, medication, or environmental influences in the relationship between specific autistic domains and BPD. Further studies could also clarify the role of trauma in the correlation between autism spectrum, BPD, and catatonia. Understanding the importance of the role of trauma in the relationship between ASD, BPD, and catatonia could also promote the inclusion, for preventive purposes, of trauma-informed care in clinical settings for individuals with BPD, especially in the presence of parallel autistic and catatonic traits. Regarding the possible limitations of the present study, it is necessary to mention how the use of self-report tools returns a less accurate assessment than the direct evaluation of a clinician. On the other hand, the use of spectrum questionnaires enabled the assessment of the examined mental disorders in the full range of their manifestations, from subthreshold symptoms to overt manifestations. Moreover, the cross-sectional design of the study does not allow us to make inferences about possible temporal or causal relationships between the disorders. It would also be important to consider the possibility of reverse causality between borderline personality disorder and catatonia but unfortunately the literature on this is still scarce. A recent study showed only that in patients with BPD and lifetime mood disorders, it is the manic/hypomanic component and not the depressive one that is correlated with various domains of the psychotic spectrum, including catatonia [37]. Finally, the study was based on a relatively limited sample: further studies in wider populations and possibly with a longitudinal design are warranted in order to clarify the relationships between autistic traits, BPD, and catatonia.

## 5. Conclusions

Individuals affected by borderline personality disorder show greater levels of catatonic and autistic spectrum manifestations compared to healthy controls and, in most cases, to patients with major depressive disorder. The CS domains “Psychomotor activity” and “Impulsivity” and the AdAS domains “Verbal Communication”, “Empathy” and “Hyper-hypo reactivity to sensory input” appear to be the ones significantly associated with the risk of presenting diagnosis of BPD.

## Figures and Tables

**Figure 1 brainsci-13-01175-f001:**
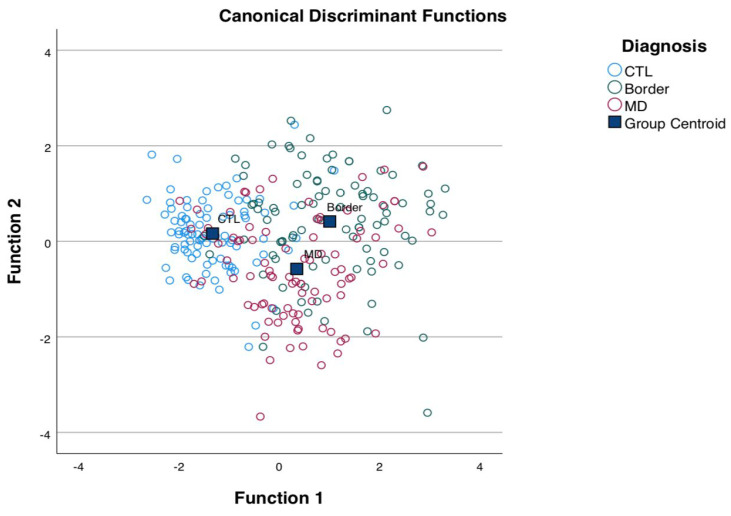
Scatter plot of the first two discriminant functions.

**Table 1 brainsci-13-01175-t001:** Comparisons of CS and AdAS Spectrum total and domain scores among the diagnostic groups.

**CS Domains**	**a. CTL** **Mean (SD)**	**b. BPD** **Mean (SD)**	**c. MDD** **Mean (SD)**	**F**	* **p** *	**Post Hoc Comparison**
Psychomotor Activity	3489 (2571)	9222 (3815)	8622 (4134)	70,077	<0.001	b > a; c > a
Verbal Response	2122 (1942)	4744 (2607)	41,333 276,901	27,867	<0.001	b > a; c > a
Repetitive Movements	1356 (1538)	2856 (2047)	2100 (1836)	15,294	<0.001	b > c > a
Artificial Expressions and Actions	0.800 (1416)	2322 (2032)	1400 (1865)	16,512	<0.001	b > a; b > c
Oppositivity or poor stimulus–response	2211 (1881)	3933 (2087)	3156 (2192)	15,812	<0.001	b > c > a
Response to Instructions	2433 (1621)	3544 (1787)	3233 (1896)	9416	<0.001	b > a; c > a
Automatisms	2544 (2380)	5411 (2587)	43,111 (2962)	26,713	<0.001	b > c > a
Impulsivity	2100 (2422)	7277 (3585)	4400 (3559)	57,888	<0.001	b > c > a
Total score	17,056 (12,167)	39,311 (15,588)	31,356 (16,533)	51,683	<0.001	b > c > a
**AdAS Spectrum domain**	**a. CTL** **(mean ± SD)**	**b. BPD** **(mean ± SD)**	**c. MDD** **(mean ± SD)**	**F**	* **p** *	**Post hoc comparison**
Childhood/Adolescence	4078 (3494)	10,344 (4850)	7789 (4948)	44,516	<0.001	b > c > a
Verbal communication	2778 (2485)	8200 (3754)	5867 (3718)	58,585	<0.001	b > c > a
Non-verbal communication	5522 (4117)	13,367 (5946)	9789 (5571)	49,960	<0.001	b > c > a
Empathy	1444 (2045)	5567 (3,0760)	4056 (3120)	50,218	<0.001	b > c > a
Inflexibility and Adherence to Routine	7944 (6034)	20,200 (8436)	14,700 (8262)	57,858	<0.001	b > c > a
Restricted Interests and Rumination	4289 (3880)	11,089 (4448)	9244 (4812)	57,564	<0.001	b > c > a
Hyper-/hyporeactivity to sensory input	1322 (1810)	6900 (4258)	5267 (4027)	58,996	<0.001	b > c > a
Total score	27,378 (19,218)	75,668 (28,216)	56,711 (29,488)	78,535	<0.001	b > c > a

**Table 2 brainsci-13-01175-t002:** Multinomial regression using CTL groups as reference categories.

Diagnostic Groups	CS Domains	B (SE)	*p*	CI (95%)
**BPD**	Psychomotor Activity	0.431 (0.116)	<0.001	1.226; 1.931
	Artificial Expressions and Actions	−0.488 (0.202)	0.016	0.413; 0.912
	Oppositivity or poor stimulus–response	−0.490 (0.190)	0.010	0.422; 0.890
	Impulsivity	0.436 (0.118)	<0.001	1.227; 1.948
	**AdAS Spectrum domains**	**B (SE)**	* **p** *	**CI (95%)**
	Verbal communication	0.304 (0.125)	0.015	1.061; 1.731
	Empathy	0.268 (0.112)	0.016	1.050; 1.627
	Hyper-/hyporeactivity to sensory input	0.312 (0.129)	0.015	1.063; 1.761
**MDD**	**CS domains**	**B (SE)**	* **p** *	**CI (95%)**
	Psychomotor Activity	0.520 (0.109)	<0.001	1.360; 2.083
	Artificial Expressions and Actions	−0.473 (0.197)	0.017	0.424; 0.917
	Oppositivity or poor stimulus–response	−0.542 (0.183)	0.003	0.406; 0.833
	**AdAS Spectrum domains**	**B (SE)**	** *p* **	**CI (95%)**
	Non-verbal communication	−0.168 (0.085)	0.048	0.716; 0.998
	Hyper-/hyporeactivity to sensory input	0.373 (0.129)	0.004	1.127; 1.869

**Table 3 brainsci-13-01175-t003:** Pooled within-groups correlations between discriminating variables and standardized canonical discriminant functions; variables are ordered by the absolute size of correlation within the function.

**CS Domains**	**Function 1**	**Function 2**
Psychomotor Activity	0.720 *	−0.307
Verbal Response	0.461 *	−0.056
Repetitive Movements	0.331 *	0.199
Artificial Expressions and Actions	0.331 *	0.299
Oppositivity or poor stimulus–response	0.341 *	0.155
Response to Instructions	0.268 *	−0.001
Automatisms	0.449 *	0.120
Impulsivity	0.633 *	0.475
**AdAS Spectrum domains**	**Function 1**	**Function 2**
Childhood/Adolescence	0.578 *	0.192
Verbal communication	0.660 *	0.260
Non-verbal communication	0.606 *	0.282
Empathy	0.617 *	0.137
Inflexibility and Adherence to Routine	0.653 *	0.291
Restricted Interests and Rumination	0.663 *	−0.017
Hyper-/hyporeactivity to sensory input	0.672 *	0.020

* Largest absolute correlation between each variable and any discriminant function.

## Data Availability

The data that support the findings of this study are available on request from the corresponding author, Amatori G. The data are not publicly available due to their containing information that could compromise the privacy of the research participants.

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
