# Peer review of "Autistic and Catatonic Spectrum Symptoms in Patients with Borderline Personality Disorder"

_brainsci, 2023, doi:10.3390/brainsci13081175_

Round 1
Reviewer 1 Report
Comments and Suggestions for Authors
The authors presented autistic and catatonic features in patients with borderline personality disorder (BPD). The paper is in the journal’s interest. The followings are my comments:
· The authors clearly presented the clinical manifestations and history of both catatonia and autism sprectrum disorder. However, possibl mechanisms between BPD and autistic traits and catatonia would be explained in more detail.
· Were any of the participants diagnosed with catatonia or autism spectrum disorder during structured interviews?
· The authors could present the statictical analyses in more detail. I would like to see how they checked the normality of the data.
· The authors discussed their findings with previous articles appropriately.
Best regards
Reviewer 2 Report
Comments and Suggestions for Authors
The article titled "Autistic and Catatonic Spectrum Symptoms in Patients with Borderline Personality Disorder" presents a valuable exploration into the connection between autistic traits (AT), borderline personality disorder (BPD), and catatonia in individuals with major depressive disorder (MDD). The study delves into the idea of an illness trajectory, suggesting that different mental disorders may align along a continuum of severity, potentially leading to the development of catatonia over time.
I have a few minor comments that may help to revise and strengthen the article.
1. While the study suggests a potential link between autistic and catatonic traits in BPD patients, it is crucial to acknowledge the possibility of reverse causation or other confounding factors that might influence these associations.
2. The discussion of specific CS and AdAS Spectrum domains related to BPD risk is insightful. However, it would be beneficial to explore potential interactions between these domains and other factors, such as comorbidities, medication, or environmental influences, which could impact the observed relationships.
3. The cross-sectional design of the study limits the ability to draw causal conclusions or establish temporal relationships between autistic traits, BPD, and catatonia. Consideration of prospective or longitudinal designs could strengthen the study's ability to establish temporal associations.
4. The article provides interesting insights into the potential role of trauma in the correlation between autism spectrum, BPD, and catatonia. To expand on this aspect, it might be valuable to discuss the implications of trauma-informed care in clinical settings for individuals with BPD and comorbid traits.
5. While the article highlights the importance of the CS and AdAS Spectrum domains related to BPD risk, it could benefit from suggesting potential implications for treatment and interventions targeting these specific domains to improve outcomes for individuals with BPD.
6. The study opens up new avenues for further research; however, it would be helpful to provide suggestions for future studies with a focus on addressing the limitations identified in the current research.
In conclusion, the article provides valuable insights into the connections between autistic traits, BPD, and catatonia. Despite the limitations, the study contributes to the existing literature and offers a foundation for future research in this complex and important area of mental health. By considering the suggested revisions and addressing the comments provided, the article can further strengthen its contributions and impact in the field.
